# Unsupervised Candidate Answer Extraction through Differentiable Masker-Reconstructor Model

**Zhuoer Wang**[1]   **Yicheng Wang**[1]   **Ziwei Zhu**[2]   **James Caverlee**[1]

[1]Texas A&M University   [2]George Mason University
{wang,wangyc,caverlee}@tamu.edu   zzhu20@gmu.edu

## Abstract

Question generation is a widely used data augmentation approach with extensive applications, and extracting qualified candidate answers from context passages is a critical step for most question generation systems. However, existing methods for candidate answer extraction are reliant on linguistic rules or annotated data that face the partial annotation issue and challenges in generalization. To overcome these limitations, we propose a novel unsupervised candidate answer extraction approach that leverages the inherent structure of context passages through a **D**ifferentiable **M**asker-**R**econstructor (**DMR**) Model with the enforcement of self-consistency for picking up salient information tokens. We curated two datasets with exhaustively-annotated answers and benchmark a comprehensive set of supervised and unsupervised candidate answer extraction methods. We demonstrate the effectiveness of the DMR model by showing its performance is superior among unsupervised methods and comparable to supervised methods. Our code and data are publicly available at https://edillower.github.io/.

## 1 Introduction

Question Generation (QG) is a burgeoning field of Natural Language Understanding and Generation. The objective of Question Generation is to produce well-structured, coherent, and valuable questions that correspond to a specific context passage and the intended answer. QG systems play a vital upstream role in enhancing the robustness and generalizability of Question Answering (QA) and Machine Reading Comprehension (MRC) models (Du et al., 2017; Dong et al., 2023), empowering chatbots and virtual assistants to answer more user needs (Gottardi et al., 2022), and powering AI-driven tutoring systems for educational purposes (Kurdi et al., 2019). For most existing QG systems, extracting qualified candidate answers from the context passage is an indispensable prerequisite to ensure that the generated questions are of high quality and relevant to the user's interests of salient information contained in the context passage.

Traditional methods for answer extraction rely on linguistic rules and models to discover the syntactic structure of the input passage's sentences. Constituency tags of Noun Phrases and Named Entity Recognition (NER) tags of person, time, location, etc., are popular choices of candidate answers. However, this kind of answer extraction method can only extract limited types of answers and usually disregards the importance of different possible answer tokens on the context and domain basis. More recently, with the development of machine learning and large language models, candidate answer extraction has been formalized as a sequence labeling problem and tackled with supervised sequence learning and classification with the use of annotated answers from MRC datasets like SQuAD (Rajpurkar et al., 2016). Nevertheless, the answer annotations of existing MRC and QA datasets have the partial annotation issue due to the annotation protocols that did not enforce exhaustive answer extraction from the context passage, leading to the overlooking of other essential details that could be beneficial in helping readers grasp the context (Bao et al., 2022). Training models with partially annotated data could result in degraded performance as the data provide misleading supervision signal (Yang et al., 2018; Zhang et al., 2017). Furthermore, the supervised learning methods still face the generalization challenge when applying them to new domains where acquiring additional human annotations is time-consuming and expensive.

To overcome the aforementioned limitations and challenges, we propose a novel unsupervised candidate answer extraction approach that leverages the inherent structure of context passage. We posit

that passage tokens can be categorized into two types – backbone tokens and information tokens. *Backbone tokens* are structural and common across various passages within the same domain, and such tokens are easily recoverable when masked. In contrast, *information tokens* are difficult to recover when masked, and such tokens are crucial information of a specific context passage, making them excellent candidate answers. In Section 3, we introduce the Differentiable Masker-Reconstructor model, in which the masker module learns to mask out tokens that the reconstructor module can then readily recover through the enforcement of self-consistency. For comprehensive assessments, we exhaustively annotated a total of 100 passages with candidate answers on SQuAD and WikiHow cooking texts as detailed in Section 4.1. We demonstrate the competitive performance of the DMR model in Section 5 by comparing it with a comprehensive list of strong baselines that cover recent advancements of both supervised and unsupervised candidate answer extraction.

In summary, we make the following contributions:

- We propose a novel Differentiable Masker-Reconstructor model, in light of recent progress on self-consistency learning and masked language models, for unsupervised candidate answer extraction.

- We release two newly created datasets with exhaustively-annotated candidate answers.

- We benchmark a comprehensive list of supervised and unsupervised candidate answer extraction methods and show the strong performance of our DMR model.

## 2 Related Work

**Candidate Answer Extraction** focuses on the extraction of salient information tokens that usually contain key information and knowledge the readers may seek from the context passage. In previous work, Named Entities are the most popular type of candidate answers (Yang et al., 2017; Lewis et al., 2019; Fabbri et al., 2020; Nie et al., 2022). In such cases, tokens of the context passage are first processed by existing NER models that tag tokens of person, organization, location, date/time, and numerical expressions. Noun Phrase (NP) is another popular choice that has been used a lot (Yang et al.,

2017; Lewis et al., 2019; Nie et al., 2022). Among them, Yang et al. and Nie et al. also consider more diverse types of phrases including Adjective Phrases (AP), Verb Phrases (VP), and sub-clauses (S) extracted from the consistency parsing models of the context passage. Specially, Nie et al. further expand recognized named entities into longer constituents for more diverse candidate answers. Despite the reliance on NER and parsing models, these linguistic and syntactic rules based candidate answer extraction methods fail to take the importance of the information into account. With the advancements of supervised learning methods for neural networks, multiple studies (Du and Cardie, 2017; Subramanian et al., 2018; Wang et al., 2019) utilize the annotated answer phrases of SQuAD, a MRC dataset, to train neural models that are capable of tagging and classifying candidate answer tokens. However, Bao et al. point out that the answer annotations of existing MRC and QA datasets have the partial annotation issue due to the annotation protocols that did not enforce exhaustive answer extraction from the context passage, leading to the overlooking of other essential details that could be beneficial in helping readers grasp the context. Training models with partially annotated data could result in degraded performance as the data provide misleading supervision signal. To address the partial annotation problem, previous work resorts to Positive-Unlabeled (PU) learning (Kiryo et al., 2017), which uses modified risk estimators to re-balance the weights of positively labeled answer tokens and the remaining deemed-as-unlabeled tokens for unbiased learning of an unbiased binary classifier. Nevertheless, as a common problem across different tasks, the supervised learning methods still face the generalization challenge when applying them to new domains where acquiring additional human annotations is time-consuming and expensive.

**Self-consistency Learning** has been adopted in the field of Natural Language Processing in recent years for tasks like text encoding (Li et al., 2015), sentence compression and summarization (Baziotis et al., 2019; Malireddy et al., 2020), NER (Iovine et al., 2022), and data-to-text generation (Wang et al., 2023). Models that leverage self-consistency learning usually include two modules that are reversals of each other. Specifically, one compressor module takes a sentence, paragraph, or document as the initial input and compresses the text into

| Article Title | Content of the step *Heat the Olive Oil* |
|:---:|:---|
| How to Make Caldo Tlalpeno | Add 2 tablespoons (30 ml) of olive oil to a large pot. Allow the oil to heat for 2 to 3 minutes on medium-high heat, or until it begins to shimmer. You can substitute vegetable or canola oil for the olive oil if you prefer. |
| How to Make Slow Cooker Spaghetti Sauce | Place a large skillet on the stove, and add 2 tablespoons (30 ml) of olive oil to it. Turn the burner to medium, and allow the oil to heat for 3 to 5 minutes or until it starts to shimmer. |
| How to Make Shrimp Bisque | Add 3 tablespoons (45 ml) of olive oil to a large pot or Dutch oven, and place it on the stove. Turn the heat to medium, and allow the oil to heat for 5 minutes, or until it starts to shimmer. If you prefer, you can substitute butter for the olive oil. |
| How to Make an Omelette in a Jar | Place a large skillet on the stove, and add 1 tablespoon (15 ml) of olive oil. Turn the heat to medium-high, and allow the oil to heat until it starts to shimmer, which should take approximately 5 minutes. You can substitute butter for the olive oil if you prefer. |
| How to Make a White Pizza | Add 2 tablespoons (30 ml) of olive oil to a medium, heavy-bottomed saucepan. Place the pan on the stove, and heat it over medium heat until it begins to shimmer, which should take approximately 5 minutes. You can substitute vegetable oil for the olive oil. |

Table 1: Content of multiple WikiHow articles with different expressions of the step *Heat the Olive Oil*

intermediate outputs of dense encoding, abstractive text, or structured data that preserve the key information of the original input text. Sequentially, another reversal module attempts to reconstruct the initial input based on the first module's intermediate condensed outputs. The two modules are progressively trained by enforcing the consistency between the initial input and reconstructed input. Notably, the differentiability of the intermediate outputs greatly affects the back-propagation of the learning signals from the reversal module to the compressor module. Except the application of self-consistency learning for text encoding, most other applications including ours have the challenge of handling non-differentiable discrete intermediate outputs. To deal with this issue, Iovine et al. and Wang et al. adopt cycle training that alternatively changes the roles of the two modules, and Baziotis et al. draw support from Gumbel-Softmax with the strike-through approach for differentiable sampling from the categorical distribution. However, the ablation study conducted by Baziotis et al. suggests that the generation of a relevant and fluent summary was mainly driven by a topic loss.

**Masked Language Modeling** (MLM) is a fundamental technique for the pretraining of Large Language Models (LLM), first popularized by the BERT model (Devlin et al., 2019). During the pretraining stage, some percentage of the input tokens are masked at random, and the model's objective is to predict the original tokens based on their context tokens. This is different from traditional language models that typically predict the next word in a sequence, enabling models to bidirectionally understand the context and achieve strong language understanding capability. Follow-

ing BERT, RoBERTa (Liu et al., 2019) also uses MLM as its pretraining objective. It improves upon BERT by using dynamic masking that changes the masking pattern applied to the input text rather than static masking that always masking the same words, which results in a more robust LLM. Along this vein, MLM becomes one of the most popular pretraining objectives for large language models, including ALBERT (Lan et al., 2020), BART (Lewis et al., 2020), Longformer (Beltagy et al., 2020), etc. that optimized BERT from different aspects.

## 3 Approach

Our approach is based on the intuition and observation that the text passages within the same domain have inherent structure shaped by the underlying writing style, knowledge space, and expression formulation. We convey these concepts by showing a typical example we sourced from the WikiHow[1] cooking articles. WikiHow is a popular online resource that offers user-curated, step-by-step guides on how to perform certain tasks. As shown in Table 1, multiple WikiHow articles on the cooking of five different dishes have the same step *Heat the Olive Oil*. We can observe explicit templates, like `Add AMOUNT of olive oil to a CONTAINER` and `Place a CONTAINER on the stove, and add AMOUNT of olive oil`, from these passages. For the step of *Heat the Olive Oil*, it is easier for the Masked Language Models to fill the masks of `[MASK] 3 tablespoons (45 ml) [MASK][MASK][MASK][MASK][MASK] large pot or Dutch oven` with correct tokens than filling the masks of `Add [MASK][MASK][MASK] of olive`

---

[1] https://www.wikihow.com/

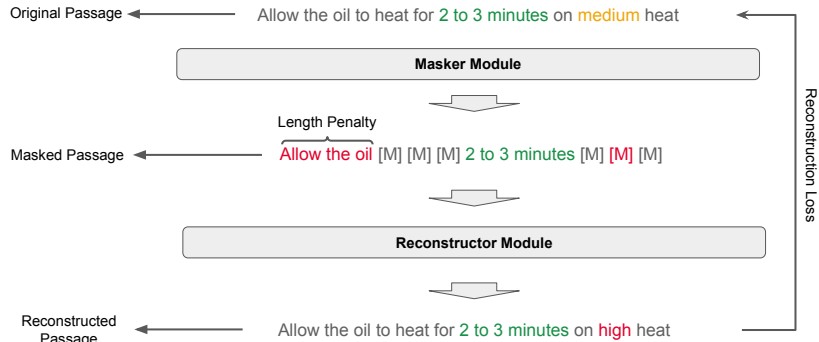

Figure 1: The Masker-Reconstructor Model (`[M]` represents the special token `[MASK]`). *Reconstruction loss* guides the learning of the Reconstructor module as well as penalizes the Masker module for masking out hard-to-recover tokens (illustrated by `[M]` and high in red). *Length penalty* enforces the learning of the Masker module so that those easy-to-recover tokens (illustrated by `Allow the oil` in red) are more likely to be masked out as masking them would yield the gain of both length and reconstruction loss.

`oil to a [MASK][MASK][MASK][MASK][MASK].` The reason is that the black tokens are repeatedly seen in the domain and result in statistically higher prediction probability when the colored tokens are given as the context for the MLM. On the other hand, the colored tokens are precise and specific information of a particular passage that do not have the prediction probability as high as the black tokens. Therefore, tokens of the text passages can be categorized into the following two types:

**Backbone Tokens** are those black tokens of the content shown in Table 1, are structural and common across various passages within the same domain, and such tokens are easily recoverable when masked.

**Information Tokens**, in contrast to Backbone Tokens, are difficult to recover when masked, and such tokens are crucial information of a specific context passage, making them excellent candidate answers. Examples of Information Tokens are colored tokens in Table 1 that express the AMOUNT, CONTAINER, TIME, HEAT-LEVEL, SUBSTITUTION of the specific context. Besides the cooking domain, Wikipedia articles regarding a person may have similar structure that has date of birth, place of birth, occupation, education, etc. as information tokens. News articles may express who, what, where, when of events as information tokens with a shared explicit or implicit template.

### 3.1 The Masker-Reconstructor Model

To extract candidate answers without accessing ground truth labels and upstream constituency parser or named entity recognizer, we develop the Masker-Reconstructor Model to discriminate backbone tokens and information tokens based on the idea of self-consistency learning as mentioned in Section 2. As illustrated in Figure 1, the Masker-Reconstructor Model consists of a Masker module and a Reconstructor module. The Masker module is a token classification model $C_M$ powered by LLM. It takes the original context passage $P$ as input and makes a binary classification for each token with 1 indicating the token should be preserved and 0 indicating the token should be masked. Based on the classification results of $C_M$ and the original passage $P$, the Masker module outputs an intermediate passage $\hat{P}$ with masked tokens represented by the special token `[MASK]`. Sequentially, the Reconstructor module, a Mask-Filling model $C_R$ powered by LLM as well, takes the intermediate masked passage $\hat{P}$ as input and predicts the conditional probability of possible original surface tokens for each `[MASK]`. Relate to other works in self-consistency learning, in our work, the Masker module and the Reconstructor module are reversals of each other, and the two modules can therefore be progressively trained by enforcing the self-consistency between the initial input and reconstructed input. Specifically, the reconstructed passage $P'$ is compared with the original passage $P$ for the calculation of the reconstruction loss as follows:

$$L(P, P') = -\frac{1}{|P|} \sum_{i=1}^{|P|} [1 - C_M(\hat{P}_i|P)]$$
$$\cdot [Prob(P_i) \cdot log(C_R(P'_i|\hat{P}))]$$

The reconstruction loss not only guides the learning of the Reconstructor module but also ensures

that the Masker module is getting more penalty if it masked out tokens that are harder for the Reconstructor module to recover. Also, this learning procedure is solely dependent on the self-consistency between the original input passage and the associated reconstructed passage. However, a short cut for the Masker module to achieve low reconstruction loss is to classify all the tokens as 1 that preserves all the tokens, which betrays our goal of finding those hard-to-recover information tokens for the candidate answer extraction purpose. To enforce the learning of the Masker module, a length penalty is needed in conjunction with the reconstruction loss. In our work, the length loss is calculated as:

$$L_{length} = \frac{1}{|P|} \sum_{i=1}^{|P|} C_M(\hat{P}_i|P)$$

Therefore, the final loss of the Masker module is:

$$L_{C_M} = L(P, P') + \lambda \cdot L_{length}$$

where $\lambda$ is a weighting factor that balances the two loss terms. Higher $\lambda$ enforces the model to mask out more tokens.

## 3.2 Differentiable Self-consistency Learning

An unaddressed challenge at this point is the non-differentiability of the intermediate outputs as the Masker's binary classification is a discrete step, which obstructs the backpropagation of the main training signal, reconstruction loss, to the Masker module. To achieve differentiable self-consistency learning, our masker-reconstructor model employs the *Straight-Through Gumbel-Softmax* estimator(Jang et al., 2017).

Gumbel-Softmax(Maddison et al., 2017), a relaxation of discrete sampling operations, introduces stochasticity into the model by generating differentiable approximations of discrete random variables. This permits the masker-reconstructor model to sample different tokens in a differentiable manner. However, standard Gumbel-Softmax does not allow for hard decisions, i.e., decisions with binary values (1 for preserved tokens, 0 for masked tokens), which are desirable in our case. Therefore, we use the *Straight-Through Gumbel-Softmax*, an estimator that offers an elegant way of approximating hard decisions while keeping the process differentiable. During the forward pass, the *Straight-Through Gumbel-Softmax* uses the Gumbel-Softmax function with temperature approaching zero, yielding

one-hot (hard) vectors. Meanwhile, in the backward pass, it can utilize the gradient calculated from the continuous approximation of the standard Gumbel-Softmax distribution.

Through this differentiable self-consistency enforced learning paradigm, we demonstrate in Section 5 that the proposed DMR model is capable of picking up information tokens that are excellent candidate answers in an unsupervised manner.

## 4 Experimental Setup

### 4.1 Data

A prevailing challenge of the candidate answer extraction task is the absence of a specific dataset for comprehensive evaluation and training. Although the answer annotations from MRC and QA datasets have been ubiquitously adapted for the task of candidate answer extraction, researchers have pointed out that these data have the partial annotation problem for the candidate answer extraction task (Bao et al., 2022). As the annotation protocols used for the construction of MRC and QA datasets did not require their annotators to find an exhaustive list of candidate answers and come up with associated questions, using data with missing annotations could provide wrong supervision signals to the candidate answer extraction models, and evaluating with this kind of data could lead to wrong conclusion. Bao et al.'s analysis found that 48.89% and 62.44% of candidate answers are missing from the SQuAD dataset and the DROP dataset (Dua et al., 2019) respectively. Our answer/context ratio analysis (available in Appendix A) on existing MRC and QA datasets also suggests that their annotated answers only cover a very small portion, with 13.35% being the highest and as low as 0.26% of the information contained in the context passages.

In consideration of the training and evaluation challenge, we prepared and curated the following three datasets to facilitate the assessment of our work:

**Original (Partially-annotated) SQuAD**: SQuAD is a large scale MRC and QA dataset with 97,095 pairs of question-answer pairs corresponding to 20,947 context passages. It is also the most frequently used dataset for candidate answer extraction in previous work. Despite the partial annotation issue, SQuAD has the highest answer/context ratio of 13.55% so far, and its partially annotated answers can still provide valuable supervision and bring some generalizability through its large scale.

| Dataset | SQuAD | WH-C |
|---|---|---|
| **Source** | Wikipedia | WikiHow |
| **Domain** | Open-domain | Cooking |
| **Context Passage Amount** | 20,947 | 16,642 |
| **Average Passage Length** | 135 Tokens | 66 Tokens |
| **Answer/Context Ratio (Original)** | 13.35% | N/A |
| **Answer/Context Ratio (Exhaustively-annotated)** | 35.11% | 50.51% |

Table 2: Dataset Statistics

We use the original SQuAD dataset mainly for the purpose of training and comparing with supervised methods, and we also report the performance of different candidate answer extraction methods on its partially annotated answers as a reference.

**Exhaustively-annotated SQuAD**: For a more comprehensive evaluation, we have exhaustively annotated a subset of SQuAD. We employed two annotators, both possessing Bachelor's degrees from accredited universities and having experience in Natural Language Processing. We followed Bao et al.'s annotation guidelines[2] to encourage annotators to label similar types of information as well as any important information missed in the original SQuAD dataset. We conducted the annotation on 50 SQuAD passages using the text annotation tool POTATO (Pei et al., 2022). The annotation resulted in a high agreement of 87.8% and a satisfactory Cohen's kappa score (Cohen, 1960; Carletta, 1996; Artstein and Poesio, 2008) of 0.7486. We combined the the two annotators agreed set of candidate answers, balancing extensiveness and importance, with the original SQuAD answers as the final labels for each passage.

**Exhaustively-annotated WH-C**: We obtained 16,142 cooking-related passages from WikiHow for the self-consistency learning. For the evaluation of candidate answer extraction on WikiHow, we annotated 50 passages following the same annotation schema as mentioned above. The annotation resulted in an agreement of 89.67% and a satisfactory Cohen's kappa score of 0.7908. As it shows in Table 2, compared to SQuAD, the WikiHow cooking data pertains to a more specialized domain with shorter context passage. However, our annotation results suggest that passages of WH-C are more information-intensive than SQuAD. This is proba-

bly due the instructional and task-oriented nature of WH-C content.

### 4.2 Baselines

We compare our approach with a comprehensive list of methods representing recent progress on both supervised and unsupervised candidate answer extraction.

**FT-LLM** leverages the recent advancement in Large Language Models (LLM) and the associated fine-tuning (FT) technique. It is a representative and robust supervised method. We train the RoBERTa classification model with the original SQuAD dataset.

**SCOPE** (Bao et al., 2022) is the state-of-the-art supervised method that attempt to learn a candidate answer extraction model from partially annotated data with the Positive-Unlabeled Learning technique. Specifically, we train the RoBERTa classification model with the Original SQuAD dataset and the PU objective function described in the SCOPE paper with prior distribution $\pi$ of all positive samples calculated from our exhaustively annotated SQuAD data.

**Noun Phrases** extracted by the off-the-shelf library spaCy (Honnibal et al., 2020).

**Named Entities** extracted by the off-the-shelf library spaCy.

**Extended NE** (Nie et al., 2022) extends each recognized named entity by finding a longer constituent that contains the named entity in the constituency parse tree with at least 80% of the sentence length.

**DiverseQA** refers to another candidate answer extraction method of Nie et al. that combines Extended NE and constituents tagged as NP, ADJP, VP, and S in the constituency parse tree as candidate answers as described in the DiverseQA paper. We also report the standalone performance of ADJP, VP, and S as the reference.

**ChatGPT** (Ouyang et al., 2022) is a recent breakthrough that has demonstrated leading zero-shot learning capability. We prompt ChatGPT [3] with the context passage accompanied by the prepended instruction *"Extracting qualified candidate answers from context passages is a critical step for most question generation systems. Please extract an exhaustive list of candidate answers (substrings from the following context passage): ......"*. For reproducible results, we set the temperature to 0 when

---

[2]Ideally, we would like to use Bao et al.'s annotation on SQuAD. However, its first author informed us that the annotated data was unfortunately lost due to a storage issue after his graduation.

[3]ChatGPT model version: `gpt-3.5-turbo` (released on March 1st, 2023).

| Dataset | Original (Partially-annotated) | | | Exhaustively-annotated | | |
|---|---|---|---|---|---|---|
| Metric | Precision | Recall | F1 | Precision | Recall | F1 |
| **Supervised Methods** | | | | | | |
| **FT-LLM** | **47.45**$_{(0.20)}$ | 33.22$_{(0.96)}$ | 39.07$_{(0.70)}$ | **60.31**$_{(3.77)}$ | 19.90$_{(1.80)}$ | 29.84$_{(1.59)}$ |
| **SCOPE** | 32.28$_{(1.65)}$ | **69.38**$_{(3.59)}$ | **44.09**$_{(0.84)}$ | 45.81$_{(0.27)}$ | **59.00**$_{(1.32)}$ | **51.57**$_{(0.49)}$ |
| **Unsupervised Methods** | | | | | | |
| **Noun Phrases** | 18.91 | 75.45 | 30.24 | 37.90 | 69.42 | 49.04 |
| **Named Entities** | **25.37** | 38.00 | 30.42 | **43.54** | 26.02 | 32.60 |
| **Extended NE** | 20.07 | 44.04 | 27.57 | 40.19 | 36.08 | 38.02 |
| **DiverseQA** | 16.87 | **94.12** | 28.62 | 35.69 | **89.26** | 50.99 |
| l- ADJP | 17.85 | 5.64 | 8.57 | 36.74 | 4.91 | 8.66 |
| l- VP | 16.23 | 47.24 | 24.16 | 30.93 | 38.84 | 34.43 |
| l- S | 15.86 | 28.22 | 20.31 | 29.01 | 24.00 | 26.27 |
| **ChatGPT** | 23.75 | 53.79 | **32.95** | 41.00 | 39.80 | 40.39 |
| **DMR** (Ours) | 18.14$_{(0.03)}$ | 80.63$_{(0.34)}$ | 29.61$_{(0.05)}$ | 37.94$_{(0.32)}$ | 77.98$_{(1.17)}$ | **51.04**$_{(0.07)}$ |

Table 3: Experiment Results on Original (Left) and Exhaustively-annotated (Right) SQuAD data. We bold the highest among supervised and unsupervised methods respectively and underline the second highest among unsupervised methods. For trained models, we report the average and standard deviation (in parenthesized subscripts) of each metric for 3 repeated runs with different random seeds.

querying ChatGPT.

## 4.3 Training Parameters

For the models trained with either supervised learning methods or the self-consistency learning method, we use the RoBERTa-base model which has 12 layers, a hidden size of 768, 12 self-attention heads, and approximately 125M parameters. We use the AdamW optimizer with linear weight decay, maximum input length of 512, learning rate of 5e-5, and effective batch size of 256. We train each model until convergence and report the average performance and standard deviation across three runs that have different random seeds. For the weighting parameter $\lambda$ of the DMR model, we initialize it as 0.35 and gradually increase it every epoch for a final value of 0.65. Data-wise, supervised learning methods have access to both context passage and annotated answers while other methods only have access to the context passage.

## 5 Results and Analysis

Table 3 displays the candidate answer extraction performance of different methods on *Original, partially-annotated, SQuAD* data and *Exhaustively-annotated SQuAD* data. Table 4 shows the counterpart on *Exhaustively-annotated WH-C* data.

The SCOPE model consistently outperforms the FT-LLM model across all three datasets. The performance gain of +5.02, +21.73, and +39.44 in F1 is mainly driven by the improvement in recall, and the performance gaps on exhaustively-annotated datasets are larger than the gap on the partially-

| Metric | Precision | Recall | F1 |
|---|---|---|---|
| **Supervised Methods** | | | |
| **FT-LLM** | **83.64**$_{(4.29)}$ | 20.65$_{(2.46)}$ | 33.03$_{(2.91)}$ |
| **SCOPE** | 67.23$_{(1.88)}$ | **78.59**$_{(1.89)}$ | **72.47**$_{(1.90)}$ |
| **Unsupervised Methods** | | | |
| **Noun Phrases** | 60.69 | 71.17 | 65.51 |
| **Named Entities** | **88.18** | 13.83 | 23.91 |
| **Extended NE** | 71.57 | 15.74 | 25.80 |
| **DiverseQA** | 52.28 | **87.65** | 65.49 |
| l- ADJP | 69.61 | 7.53 | 13.58 |
| l- VP | 46.42 | 53.58 | 49.74 |
| l- S | 47.76 | 33.28 | 39.23 |
| **ChatGPT** | 75.91 | 52.78 | 62.27 |
| **DMR** (Ours) | 58.20$_{(1.04)}$ | 78.04$_{(2.73)}$ | **66.64**$_{(0.48)}$ |

Table 4: Experiment Results on Exhaustively-annotated WH-C data (format is the same as Table 3).

annotated dataset. These results signify the partial annotation problem that brings misleading training signal and evaluation challenge. The high precision of Named Entities suggests that tokens of person, organization, location, date/time, and numerical expressions are key pieces of information in a passage that the reader might seek. However, the low recall implies that key information can also be expressed in various forms beyond named entities. The high recall of Noun Phrases reflects the dominant role of noun words in conveying information, but the low precision hints at the difference in information importance from the reader's perspective. Also, it is noteworthy that the recall of Noun Phrases dropped by 6.03 basis points when evaluation on SQuAD data changed to the exhaustively-annotated version. This drop suggests that many complex and diverse

forms of candidate answers have been omitted by the crowd-sourcing workers of SQuAD.

For the partially-annotated data, as the precision of each method might be underestimated due to the partial annotation of answers, we mainly look at the recall metric to assess their coverage of the existing answers. Methods that intend to extract a more comprehensive set of candidate answers including SCOPE, DMR, and DiverseQA have achieved high recall as expected. As for the evaluation on exhaustively-annotated data, the aforementioned methods were able to maintain high recall. Notably, SCOPE consistently achieved the best performance, as measured by F1, with the help of a mass amount of annotated answers from the original SQuAD dataset. When such annotated data is not available, our DMR model achieved the best performance among the unsupervised methods. Compared to DiverseQA, another competitive unsupervised method, the performance gain of DMR is mainly accomplished by balancing the precision, which emphasizes the importance of potential candidate answers, and recall, which emphasizes the extensiveness and diversity of potential candidate answers.

We also would like to point out that, compared to unsupervised methods with second highest F1, our DMR model has the performance gain of +0.05 and +1.13 on exhaustively-annotated SQuAD and WH-C respectively. The gain on the former dataset is quite insignificant while substantial on the later dataset. We attribute this difference to the fact that WH-C is more focused on a specialized cooking domain while SQuAD consists of open-domain data. Therefore, WH-C should have a more prominent underlying structure that can be captured by the DMR model through self-consistency learning to discriminate backbone tokens and information tokens. This finding could guide the future development and application of DMR model.

Lastly, it is also interesting to see that ChatGPT achieved the second highest precision across all the datasets among unsupervised methods. Upon checking ChatGPT extracted answers, we find ChatGPT significantly prefers to extract Noun Phrases and Named Entities as candidate answers. This preference results in the omission of more diverse answer forms and a lower recall.

## 6   Conclusions

In this work, we target the underexplored task of unsupervised candidate answer extraction and introduce a novel Differentiable Masker-Reconstructor model inspired by recent progress in self-consistency learning and masked language models. We have created two exhaustively annotated datasets that mitigate the partial annotation issue of existing MRC/QA datasets, thus allowing for a fairer evaluation of candidate answer extraction methods. We demonstrate that the DMR model, without the reliance on annotated data and external tools, achieves superior performance compared to other unsupervised methods, and its performance is also comparable to that of supervised methods.

## Limitations and Future Work

Despite the effort we have made, our work still has the following limitations:

**Relatively Low Precision**: Although the DMR model is capable of extracting extensive and diverse candidate answers, the precision of our model has a large gap with the supervised methods as well as Named Entities based methods. Part of the reason is that the reconstructor model usually relies on the first word of each sentence and sometimes few other structural words for a success reconstruction of the original passage. Such tokens are not valuable candidate answers and could add noise to the extractions. Hence, it's necessary to explore alternative ways to better encode the underlying structure beyond using preserved tokens. Also, given the fact that partially annotated answers are available for some domains. It worth exploring possible ways of incorporating those valuable labels with the DMR model.

**Fragmented Candidate Answer Phrases**: The DMR model sometimes would mask out structural tokens within a candidate answer phrase. For instance, it may mask out the token "to" for the phrase "3 to 5 minutes", which results in fragmented answers. As a future work, we would like to explore the possibilities and approaches of masking and reconstructing by constituents instead of doing it by tokens.

**Effect on downstream applications**: The effectiveness of different candidate answer extraction models on downstream applications can be affected by various factors including candidate answer refining methods, capabilities of question generation models, capabilities of question answering mod-

els, etc. Due to the space and resource constraints, we were not able to conduct comprehensive experiments and analysis to quantify the effect of different candidate answer extraction methods on downstream applications. In the future, we would like to explore if the DMR model itself can be used as a refining method in collaboration with other extraction methods, and we also would like to design controlled experiments to see the effect of different candidate answer extraction methods on downstream QG and QA tasks.

## Ethics Statement

Although our DMR model has achieved strong performance, it may still extract undesired or incorrect tokens as candidate answers. The errors generated by our model can propagate and amplify in downstream models, such as those used in QG and QA systems. As a result, QG systems may generate inappropriate or offensive answers and QA systems may provide inaccurate responses or misleading information, which could lead to real-world consequences. We urge downstream users and end users to use our models and data with caution.

## Acknowledgements

We would like to express our gratitude to the three reviewers and the meta reviewer for their valuable suggestions.

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

# Appendix

## A   Analysis of Extractive MRC/QA Datasets

In this section, we analyze extractive MRC/QA datasets that has been used at the MRQA 2019 Shared Task (Fisch et al., 2019). Our analysis is based on the preprocessed data released by the MRQA 2019 organizer, and the detailed statistics and comparison can be found in Table 5 (continued in next page).

| Dataset | Question/Answer Source | Context Source | Context Token Length | Anser/Context Ratio |
|---|---|---|---|---|
| SQuAD | Crowdsourced | Wikipedia | 134.88 | 13.35% |
| NewsQA | Crowdsourced | News Articles | 589.82 | 6.69% |
| TriviaQA | Trivia | Web Snippets | 753.42 | 0.31% |
| SearchQA | Jeopardy | Web Snippets | 738.57 | 0.26% |
| HotpotQA | Crowdsourced | Wikipedia | 176.84 | 1.80% |
| Natural Questions | Search Logs | Wikipedia | 205.03 | 7.61% |
| BioASQ | Domain Experts | Science Articles | 241.29 | 1.23% |
| DROP | Crowdsourced | Wikipedia | 225.66 | 7.75% |
| DuoRC | Crowdsourced | Movie Plots | 692.14 | 2.33% |
| RACE | Domain Experts | Examinations | 343.42 | 0.93% |
| Relation Extraction | Synthetic | Wikipedia | 29.25 | 11.06% |
| TextbookQA | Domain Experts | Textbook | 571.21 | 1.28% |

Table 5: Statistics and comparison of MRC/QA datasets.