# OpenReview forum: "Unsupervised Candidate Answer Extraction through Differentiable Masker-Reconstructor Model"
_EMNLP/2023/Conference — EMNLP 2023 Findings_

### Official Review · Reviewer_P2SC · 2023-08-04

**Soundness:** 3

**Excitement:**

3: Ambivalent: It has merits (e.g., it reports state-of-the-art results, the idea is nice), but there are key weaknesses (e.g., it describes incremental work), and it can significantly benefit from another round of revision. However, I won't object to accepting it if my co-reviewers champion it.

**Paper Topic And Main Contributions:**

In this work, the authors tend to tackle the unsupervised candidate answer extraction for QA(question answering) though DMR(Differentiable Masker-Reconstructor) model. The proposed model enforce of self-consistency for picking up salient information tokens to overcome the limitations of existing NER methods, which can only extract a limited number of types, and deep learning methods, which do not work well on partial annotation datasets. The proposed DMR model classifies the text token to two types: (a) backbone tokens and (b) information tokens. (a) The backbone tokens are readily recoverable because of structural and common tokens within the same domain. (b) On the other hand, information tokens are difficult to recover when masked because it have crucial information of a specific context passage. In addition, the proposed methods are verified by supervised/un-supervised experiment with SQuAD dataset and WH-C dataset. Furthermore, authors release two newly created datasets with exhaustively-annotated candidate answers.

**Questions For The Authors:**

1. Is it possible that there are cases where the token is readily to recover, but it contains core information? For example, if the question is about common sense, it seems that the information is likely to be important, although it is easy to recover.

2. Does the length penalty actually work to prevent masking the first n tokens of a sentence? According to the equation, it seems to operate on the entire sentence instead of forcing the first n tokens of the sentence.

3. In order to train the Masker Module, is it  utilize a pre-annotated dataset in supervised manner or to have the model itself determine the tokens to be masked in an unsupervised manner?

4. What is the meaning of P_i in the equation in section 3.1? In this paper, P_i generally means the token, but in this equation, it seems like the probability. It can cause confusion.

**Reasons To Accept:**

1. Well written and easy-to-read paper
2. Exploring in candidate answer extraction with differential masker-reconstructor model
3. Newly released exhaustively-annotated dataset

**Reasons To Reject:**

1. The meaning of the word in the equation is not clear. In this paper, P_i generally means the token, but in section 3.1's equation, it seems like the probability.

2. It is not clear whether the length penalty actually works for the first n tokens. According to the equation, it seems to be more influential in determining the masking ratio throughout the entire sentence.

3. It is not clear whether the exhaustively-annotated data actually helps QA. Instead of carrying out the entire QA framework, the authors declared that they only focused on the answer candidate selection. However, it seems necessary to verify at least the groundtruth of the exhaustively-annotated data set. This is because the more answer candidate tokens exist, the more likely the performance of the QA is to be worse. Rather than omitting the answer token, it seems necessary to have grounds to support the authors' argument that extracting more answer candidate tokens, which contain answer, is beneficial for QA.

**Reproducibility:**

4: Could mostly reproduce the results, but there may be some variation because of sample variance or minor variations in their interpretation of the protocol or method.

**Reviewer Confidence:**

4: Quite sure. I tried to check the important points carefully. It's unlikely, though conceivable, that I missed something that should affect my ratings.

---

> ### Author Rebuttal · Authors · 2023-08-28
>
> We appreciate reviewer P2SC's in-depth review and constructive suggestions. We also thank the reviewer’s acknowledgement of our *exploration* and the *contribution of new datasets*.
>
> For Reasons To Reject 1 and Question For The Authors 4, we thank the reviewer for pointing out the potential ambiguities of P_i. In the loss functions, P_i means the probability of the i-th token of the Passage. We note that the shared P initials of Probability and Passage may cause confusions, and we will use more distinctive symbols and add some additional textual explanations in the revised version of our paper.
>
> For Reasons To Reject 2 and Question For The Authors 2, if we understand it correctly, the reviewer refers to the illustration we were given in Figure 1. The length penalty does affect the masking ratio of the entire sentence jointly with the reconstruction loss. Without the length penalty, the masker module could take the shortcut of not masking any tokens or very few tokens to achieve low reconstruction loss as we have mentioned in lines 316~321 of our paper. When the length penalty is enforced, the masker model is more likely to mask out those successfully recovered tokens, as we have illustrated in the figure, since masking this kind of token would yield the gain of both length and reconstruction loss. We will add an explaining caption for Figure 1 to make these points more clear.
>
> For Reasons To Reject 3, we would like to first clarify that our work is targeting the task of extracting candidate answers mainly for QG systems. Such candidate answers naturally exist in the text passage regardless of the existence or performance of any QA systems. As for the evaluation on downstream applications, we would like to highlight that our study targets an understudied part of the existing Question Generation framework. Although augmenting data for QA is a major use case of QG, QG also has a much broader impact on building chatbots, virtual assistants, and educational systems as we have noted in lines 32~39 of the paper. We recognize the significance of the study on our approach's effect on downstream applications like QA. However, take QA as an example, many factors including but not limited to the generation capability of the QG system and the comprehension capability of the QA system would affect the final QA performance. The fair assessment of different candidate answer extraction approaches' effectiveness based on downstream tasks would require careful design of a large set of experiments to test different QG and QA models to attribute, differentiate, and quantify the impacts of each factor on the final QA performance, which we believe is a different scope and can formalize a separate full paper. To ensure our work has an adequate focus and effective evaluation, we opt to test and compare different methods directly on the target task of candidate answer extraction on the ground-truth annotation. A similar scenario in the field of NLP is that POS tagging has been a fundamental task that could be incorporated and benefit many downstream applications. However, the effectiveness of a newly proposed POS tagging method may only be directly evaluated with the task's ground-truth annotation instead of a full stack of downstream tasks.
>
> For Question For The Authors 1, we acknowledge the existence of the reverse cases, however the general pattern aligns with our observation on the different recovering difficulties of backbone tokens and information tokens. For the preservation of common sense related tokens, it depends on if it has been reiterated a lot in the corpus of the target domain to formalize an underlying structure. An example is the corpus of WikiPedia, although it contains much commonsense knowledge, most of such knowledge is only elaborated very few times compared to those structural expressions under the background of the huge text space.
>
> For Question For The Authors 3, the Masker module does NOT rely on any pre-annotated datasets in a supervised manner. It learns from the enforcement of self-consistency (reconstruction loss) and the length penalty.
>
> Thanks again for your time and valuable suggestions. We hope our response could address your concerns.

---

### Official Review · Reviewer_HZLu · 2023-08-04

**Soundness:** 3

**Excitement:**

3: Ambivalent: It has merits (e.g., it reports state-of-the-art results, the idea is nice), but there are key weaknesses (e.g., it describes incremental work), and it can significantly benefit from another round of revision. However, I won't object to accepting it if my co-reviewers champion it.

**Paper Topic And Main Contributions:**

This paper proposes a new approach for the answer candidate extraction task. This task aims to extract answer candidates from a passage. Existing works rely on a rule-based approach, NER, or a supervised sequence tagger. The authors propose a Masker-Reconstructor Model, which jointly learns the "masking model" and "reconstruction model". The masking model decides which token to be masked and the reconstruction model tries to reproduce the original unmasked passage. If the masking model masks tokens that cannot be reconstructed from the context in a passage, the mask model gets a penalty so the proposed model is intended to properly mask tokens that can be inferred from the contexts.

The evaluation was conducted on several datasets including newly curated ones. The proposed method achieves better F1 scores.

The main contributions are:
- this paper proposes an interesting unsupervised approach for answer candidate extractions.
- the proposed model achieves better performance in terms of F1 score.
- the authors will release new datasets for the answer candidate generation task.

The proposed model is reasonable and the paper is very well-written.

However, the answer extraction task is quite narrow task and mainly a subpart of data augmentation techniques for improving downstream QA tasks, i.e., extracted answers are then used for the question generation module for obtaining synthetic QA pairs for augmentation. Thus, the experiments to check the effects on the downstream QA task are necessary for this type of study (This is also mentioned in the limitation section by the authors.).

From the results in Table 3, the F1 score gained only 0.07 points (from 50.99 to 51.04) when the authors compared it with an existing best-performing model. I am interested in such a rather small gain attribute to the downstream QA tasks.

**Questions For The Authors:**

Is there any example of answer candidates that cannot be extracted by NER (or other existing methods) but can be extracted the proposed model? I could not immediately think of a case where a token other than a named entity could be an answer candidate because most gold answer candidates may be NEs.

**Reasons To Accept:**

- this paper proposes a interesting unsupervised approach for answer candidate span extractions.
- the proposed model achieves better performance in terms of F1 score.
- the authors will release a new datasets for the answer candidate generation task.

**Reasons To Reject:**

- lack of experiments on downstream QA tasks.

**Reproducibility:**

4: Could mostly reproduce the results, but there may be some variation because of sample variance or minor variations in their interpretation of the protocol or method.

**Reviewer Confidence:**

3: Pretty sure, but there's a chance I missed something. Although I have a good feel for this area in general, I did not carefully check the paper's details, e.g., the math, experimental design, or novelty.

---

> ### Author Rebuttal · Authors · 2023-08-28
>
> We appreciate reviewer HZLu's in-depth review and are delighted to see that the reviewer deems our proposed method as an *interesting approach with better performance* and acknowledges the *new datasets* we contributed.
>
> Regarding the reviewer's concerns, we would like to highlight that our study targets an understudied part of the existing Question Generation framework. Although augmenting data for QA is a major use case of QG, QG also has a much broader impact on building chatbots, virtual assistants, and educational systems as we have noted in lines 32~39 of the paper. We recognize the significance of the study on our approach's effect on downstream applications like QA. However, take QA as an example, many factors including but not limited to the generation capability of the QG system and the comprehension capability of the QA system would affect the final QA performance. The fair assessment of different candidate answer extraction approaches' effectiveness based on downstream tasks would require careful design of a large set of experiments to test different QG and QA models to attribute, differentiate, and quantify the impacts of each factor on the final QA performance, which we believe is a different scope and can formalize a separate full paper. To ensure our work has an adequate focus and effective evaluation, we opt to test and compare different methods directly on the target task of candidate answer extraction on the ground-truth annotation. A similar scenario in the field of NLP is that POS tagging has been a fundamental task that could be incorporated and benefit many downstream applications. However, the effectiveness of a newly proposed POS tagging method may only be directly evaluated with the task's ground-truth annotation instead of a full stack of downstream tasks.
>
> As for the small gain of F1 over existing methods on SQuAD, we would like to emphasize that although the performance gain varies on different datasets (+0.07 on SQuAD, +1.13 on WH-C compared to the best-performing unsupervised methods), our method is capable of adapting to the new domains through unsupervised learning from the unlabeled text that leverages domain-specific inherent structure, which empowers it to constantly achieve competitive performance that balances the extensiveness and importance of the extracted candidate answers.
>
>
> *Is there any example of answer candidates that cannot be extracted by NER (or other existing methods) but can be extracted the proposed model? I could not immediately think of a case where a token other than a named entity could be an answer candidate because most gold answer candidates may be NEs.*
>
> As you can see from the recall of Table 3 and Table 4, Named Entities constitute 38% of the ground-truth candidate answer tokens in Original partially-annotated SQuAD, 26.02% of Exhaustively-annotated SQuAD, and 13.83% of Exhaustively-annotated WH-C. For Noun Phrases (NP covers most NE), those numbers are 75.45, 69.42, and 71.17 respectively. Accordingly, there still exists a large number of answer candidates that cannot be extracted by NER.
>
> For example, in *"The fact that chloroplasts are surrounded by a double membrane is often cited as evidence that they are the descendants of endosymbiotic cyanobacteria."*, (*chloroplasts are*) *surrounded by a double membrane* could be a good candidate answer as it contains key information regarding what is often cited as evidence of them as descendants of endosymbiotic cyanobacteria. This cannot be extracted by the NER/NP, but our method could capture it as the information is very context-specific and uncommon across different passages. Although DiverseQA may capture this candidate answer by incorporating VP, it also extracts structural, unimportant phrases like "The fact" as a standalone candidate answer as the method considers all those different kinds of phrases regardless of their importance in the context while our method adaptively achieved a better balance of the extensiveness and importance as mentioned before.
>
> Thanks again for your time and valuable suggestions. We hope our response could address your concerns.

---

### Official Review · Reviewer_wx9Z · 2023-08-04

**Soundness:** 3

**Excitement:**

3: Ambivalent: It has merits (e.g., it reports state-of-the-art results, the idea is nice), but there are key weaknesses (e.g., it describes incremental work), and it can significantly benefit from another round of revision. However, I won't object to accepting it if my co-reviewers champion it.

**Paper Topic And Main Contributions:**

According to the partial labeling problem and generalization challenge faced by the existing candidate answer extraction methods for language rules or unlabeled data, this paper proposes a novel unsupervised candidate answer extraction method ( DMR ).This method utilizes the inherent structure of the context paragraph, uses the  Differentiable Masker-Reconstructor Model to enhance self-consistency to extract key information, and release two newly created datasets with exhaustively-annotated candidate answers. In general, the research motivation of this paper is clear, the research status is relatively sufficient, and it has good innovation.

**Reasons To Accept:**

1. The method proposed in this paper uses the inherent structure of context paragraphs to divide words ( tokens ) into structural words ( backbone tokens ) and information words ( information tokens ). It combines self-consistency learning with mask language modeling, the mask module shields words, and the reconstruction module can recover words through self-consistency to extract key information in an unsupervised way.
2. Two annotation datasets are created to mitigate the partial annotation issue existing MRC/QA datasets.
3. This structure of the paper is rational and clear writing.
4. The experiment of the paper is sufficient, and the validity of the proposed model is verified by comparing supervised and unsupervised methods.

**Reasons To Reject:**

The mechanism of this paper is not indepth and needs to be further elaborated.

**Reproducibility:**

4: Could mostly reproduce the results, but there may be some variation because of sample variance or minor variations in their interpretation of the protocol or method.

**Reviewer Confidence:**

3: Pretty sure, but there's a chance I missed something. Although I have a good feel for this area in general, I did not carefully check the paper's details, e.g., the math, experimental design, or novelty.

---

> ### Author Rebuttal · Authors · 2023-08-28
>
> We appreciate reviewer wx9Z's in-depth review and the recognition of our paper's *clear research motivation*, *good innovation*, and *verified validity with sufficient experiments*.
>
> We acknowledge the reviewer's concern regarding the elaboration of the mechanism. We intended to maintain a balance between brevity and clarity. However, we understand the importance of ensuring the potential audience of that paper could fully understand the mechanism of our approach. In the revised version, we will expand Section 3 to include more elaboration on the mechanism of self-consistency learning in our context and add associated pointers to some of the related work we have mentioned in Section 2.
>
> Thanks again for your time and valuable suggestions. We hope our response could address your concerns.

---

### Meta-Review · Area_Chair_EXBU · 2023-09-24

**Recommendation:** 4

**Metareview:**

This paper is focused on the topic of unsupervised candidate answer extraction for QA. It introduces a novel approach using the DMR model. The reviewers largely appreciate the paper's innovation, clarity, and contributions (the DMR model and annotated datasets). The reviewers offered many constructive comments, and there were major questions regarding better understanding the experimental results and the significance of the proposed approach on downstream QA tasks. Overall this is an interesting work that is worth sharing with the community.

---

### Decision · Program_Chairs · 2023-10-07

**Decision:**

Accept-Findings

**Comment:**

This paper is focused on the topic of unsupervised candidate answer extraction for QA. It introduces a novel approach using the DMR model. The reviewers largely appreciate the paper's innovation, clarity, and contributions (the DMR model and annotated datasets). The reviewers offered many constructive comments, and there were major questions regarding better understanding the experimental results and the significance of the proposed approach on downstream QA tasks. Overall this is an interesting work that is worth sharing with the community.